# Dating Violence Victimization among Adolescents in Europe: Baseline Results from the Lights4Violence Project

**DOI:** 10.3390/ijerph18041414

**Published:** 2021-02-03

**Authors:** Carmen Vives-Cases, Belén Sanz-Barbero, Alba Ayala, Vanesa Pérez-Martínez, Miriam Sánchez-SanSegundo, Sylwia Jaskulska, Ana Sofia Antunes das Neves, Maria João Forjaz, Jacek Pyżalski, Nic Bowes, Dália Costa, Katarzyna Waszyńska, Barbara Jankowiak, Veronica Mocanu, María Carmen Davó-Blanes

**Affiliations:** 1Department of Community Nursing, Preventive Medicine and Public Health and History of Science, University of Alicante, 03690 Alicante, Spain; carmen.vives@ua.es (C.V.-C.); vanesa.perez@ua.es (V.P.-M.); mdavo@ua.es (M.C.D.-B.); 2CIBER of Epidemiology and Public Health (CIBERESP), 28029 Madrid, Spain; 3National School of Public Health, Carlos III Institute of Health, 28029 Madrid, Spain; jforjaz@isciii.es; 4Department of Statistics, University Carlos III of Madrid, 28903 Madrid, Spain; aayala@est-econ.uc3m.es; 5Research Network on Health Services for Chronic Diseases (REDISSEC), 28029 Madrid, Spain; 6Department of Psychology of Health, University of Alicante, 03690 Alicante, Spain; miriam.sanchez@ua.es; 7Faculty of Educational Studies, Adam Mickiewicz University, 61-712 Poznan, Poland; sylwia.jaskulska@amu.edu.pl (S.J.); pyzalski@amu.edu.pl (J.P.); katarzyna.waszynska@amu.edu.pl (K.W.); barbara.jankowiak@amu.edu.pl (B.J.); 8Department of Social and Behavioural Sciences, Institute University of Maia, 4475-690 Maia, Portugal; asneves@ismai.pt (A.S.A.d.N.); daliacosta@iscsp.utl.pt (D.C.); 9Department of Applied Psychology, Cardiff Metropolitan University, Cardiff CF52YB, UK; nbowes@cardiffmet.ac.uk; 10Faculty of Medicine, Grigore T. Popa University of Medicine and Pharmacy, 700115 Iasi, Romania; veronica.mocanu@umfiasi.ro

**Keywords:** dating violence, adolescents, risk factors, machismo, acceptance of violence, sexism

## Abstract

Dating violence (DV) among adolescents is a public health issue because of its negative health consequences. In this study, we aimed to analyse the prevalence and the psychosocial and socioeconomic risk and protective factors associated DV among male and female adolescents in Europe. It was performed a cross-sectional study based on a non-probabilistic sample of 1555 students aged 13–16 years (2018–2019). The global prevalence of DV victimization was significantly greater among girls than boys (girls: 34.1%, boys: 26.7%; *p* = 0.012). The prevalence of DV in both girls and boys was greater for those over age 15 (girls: 48.5% *p* < 0.001; boys: 35.9%; *p* = 0.019). There was an increased likelihood of DV victimization among girls whose fathers did not have paid employment (*p* = 0.024), who suffered abuse in childhood, and reported higher Benevolent Sexism [PR (CI 95%): 1.01 (1.00–1.03)] and machismo [1.02 (1.00–1.05)]. In the case of boys, the likelihood of DV increased with abuse in childhood (*p* = 0.018), lower parental support [0.97 (0.96–0.99)], high hostile sexism scores (*p* = 0.019), lower acceptance of violence (*p* = 0.009) and high machismo (*p* < 0.001). Abuse in childhood was shown to be the main factor associated with being a victim of DV in both population groups, as well as sexism and machismo attitudes. These results may contribute to future DV prevention school programs for both, teenagers and children of elementary school ages.

## 1. Introduction

Dating violence (DV), defined as the commission of violence by one or both members of a couple in the context of dating, is a serious public health problem due to its alarming prevalence and damaging effects on the health and wellbeing of young teenage boys and girls. According to the European Violence against Women Survey (2013), the prevalence of currently physical and/or sexual intimate partner violence (IPV) among young women ages 18–29 is 6% and 48% in the case of lifetime psychological IPV. In contrast, the registered prevalence among adult women over age 30 is around 4% and 32%, respectively [1]. In other countries, current physical and psychological IPV among women ages 15–24 is estimated to be around 30% [2]. Similarly, it has been observed that male adolescents also suffer from DV, though the dynamics and consequences may be more severe for girls [3].

Dating violence has a wide range of consequences for youth and adolescent populations, including an increased likelihood of other violence-related behaviour, substance use, depression, suicidal thoughts, poorer educational outcomes, post-traumatic stress, physical health and risky sexual behaviour [3,4]. DV also has long-term consequences for youth development. It affects how young people learn to cope with difficult situations and can lead to reduced academic performance, which negatively impacts their health and wellbeing [5]. Moreover, DV has been considered a potential gateway to adult intimate partner violence later in life [6].

In recent decades, several prevention programmes have been developed in order to reduce DV [7,8]. Despite of this fact, rates of DV remain high, suggesting that a more integrative and critical approach is needed, focused on the multidimensionality of DV phenomena [9]. According to the positive youth development perspective [10], youth strengths can be used as protective factors to cope with interpersonal violence. In contrast to traditional deficit-focused perspectives, a strength-based approach emphasizes youth agency and the capability to use internal and external assets for effective personal and social functioning. Internal assets are perceived as a person’s positive competencies (e.g., empathy, communication skills, self-esteem), while external assets refer to positive resources provided by the social and community support network (such as empowerment, parental monitoring, positive peer influence) [11]. Related to the positive development approach, it has been noted that school-based interventions can be a promising approach to enhance success in school and increase interpersonal skills in order to reduce DV. However, scientific research on DV protective factors is still scarce, and a more comprehensive understanding of the potential of this approach potential is still needed.

In previous research on potential stressors related to dating violence, there has been a tendency to focus separately on adolescents’ social circumstances, such as socioeconomic conditions or ethnic background; exposure to other forms of violence, such as prior or current family violence; lack of social support; parental style; and the presence of harmful personal skills such as aggressivity [3,12,13]. Dating violence is structural by nature and has been also associated with traditional gender roles and previous exposure to family violence [14]. In fact, the acceptance of sexism and machismo seems to be linked to a set of social representations that legitimizes the use of violence in intimate relationships, including among youth [15]. There is still a weak evidence base concerning adolescents in Europe in this issue [16,17]. There is also a need for studies that integrate the wide variety of potential precursors and protective factors of DV, which may contribute to public health strategies that prevent DV and promote healthy relationships. Also, the European Gender Equality Strategy 2020–2025 [18] could benefit from studies that provide cross-country results.

We conducted a European educational project “Lights, Camera and Action against Dating Violence” (Lights4Violence) during the period 2017–2019. The project was funded by the European Commission, Directorate-General for Justice and Consumers Rights, Equality and Citizen Violence Against Women Program of 2016. As a part of the project, we carried out an educational program to promote personal and external assets to promote healthy relationships among adolescents from different European cities (Alicante, Rome, Iasi, Poznan, Matosinhos and Cardiff) [19].

This study aims to analyse the prevalence of psychosocial and socioeconomic risk and protective factors associated with being a victim of dating violence (DV) among male and female adolescents in Europe who participated in the Lights4Violence project baseline.

## 2. Materials and Methods

This study used a cross-sectional design. The data was collected using an online questionnaire distributed to the schools of each country during the 2018–2019 academic year.

### 2.1. Participants

We recruited a non-probabilistic sample of 1555 high school students ages 13–16 in Alicante, Spain (*n* = 255, 50.98% girls); Rome, Italy (*n* = 285, 72.28% girls); Iasi, Romania (*n* = 343, 62.39% girls); Matosinhos, Portugal (*n* = 259, 48.26% girls); Poznan, Poland (*n* = 190, 71.05% girls) and Cardiff, UK (*n* = 204, 54.90% girls). The mean age of our sample was 14.34 years old (std: 1.47) (girls: 14.4, sdt: 1.42; boys: 14.2, sdt: 1.44). The 47.3% of adolescents (47.4% of the girls and 46.1% of the boys) were between 14–15 years old.

School selection was carried out by contacting different secondary education centers from the city as considered appropriate by the members of the research team (non-random sample). A statistical power analysis was performed for sample size estimation based on data from a previously published, random-effects meta-analysis of 23 studies concerning school-based interventions that aimed to prevent violence and negative attitudes in teen dating relationships [20]. The initial sample was designed for 1300 students.

Data was gathered in two schools per country (total of 12 schools) between October 2018 and February 2019. The percentage of participation was 98.78%. For this study, we selected students who reported having been involved in a romantic or dating relationship, or were considering a personal relationship, in person and also by distance, through letters or via the Internet. Students who declared that their gender identity was ‘other’ (not identifying as male or female) were excluded from the final sample of this study (1.75%).

### 2.2. Main Outcome

To measure exposure to dating violence, those who had ever been in a dating relationship were asked: “Has anyone that you have ever been on a date with physically hurt you in any way? (For example, slapped you, kicked you, pushed, grabbed, or shoved you)”; “Has a person that you have been on a date with ever attempted to force you to take part in any form of sexual activity when you did not want to?”; “Has a person that you have been on a date with ever threatened you or made you feel unsafe in any way?”; “Has a person that you have been on a date with ever tried to control your daily activities, for example, whom could you talk with, where you could go, how to dress, check your mobile phone etc.?”. Exposure to intimate partner relationships and, among these, to dating violence, was measured by a variable created for the data analysis with the following categories: has never been in a partner relationship; has been in a relationship but never experienced violence; has been in a relationship and has experienced violence (physical and/or sexual and/or fear and/or control) [21].

### 2.3. Covariates

Sex, age, and father’s employment were used in two categories: paid work (paid work and self-employed) and non-paid work (exclusively dedicated to the home, unemployed, retired/unable to work because of a disability, student, died or do not know).

Experiences of abuse and/or violence by an adult in childhood before 15 years of age [21]. Three questions with dichotomous answers (yes/no) were included: “Before you were 15 years old, did any adult -that is, someone 18 years or older- physically hurt you in any way? (For example, slapped you, kicked you, pushed, grabbed, or shoved you)”; “Before you were 15 years old, did someone 18 years or older force you to participate in any form of sexual activity when you did not want to?”; “Before you were 15 years old, did you witness in your family environment someone (your father or your mother’s partner) physically beat or mistreat your mother?”.

Perceived social support was measured through the Student Social Support Scale (SSSS) [22]. The scale is made up of five subscales with 12 items each: parents, teachers, classmates, close friends and school staff. Each of the items is answered in a scale with 6 Likert-type categories that indicate the level of support received (never, almost never, some of the time, most of the time, almost always and always).

For sexism, we used the Ambivalent Sexism Inventory [23]. This scale is composed of 22 items that measure the level of agreement on a scale with six Likert-type categories, ranging from *totally disagree* to *totally agree*. The scale presents two subscales with 11 items each: benevolent sexism (BS) and hostile sexism (HS).

Violent thinking was measured using the Maudsley Violence Questionnaire (MVQ) [24]. This questionnaire presents 56 items on a dichotomous scale of true and false, with sentences that justify and support violence. It is made up of two subscales: acceptance (14 items) and machismo (42 items).

Capacity for problem resolution was measured using the Social Problem-Solving Inventory-Revised Scale (SPSI-R) [25]. It is made up of 25 items related to important problems that cause worry and cannot be immediately resolved. Its response scale consists of five categories of Likert-type responses, ranging from *not at all certain* to *extremely certain*.

### 2.4. Data Analyses

We carried out descriptive analyses (averages and percentages) of the sample for each of the variables used in the study. Prevalences were calculated for dating violence and each of the different types of violence. We analysed the differences in dating violence for each of the covariates using chi-square tests (qualitative variables) and *t*-tests for students (quantitative variables). In order to understand which variables were associated with dating violence, we calculated prevalence ratios (PR) using Poisson regression with robust variance. Statistical significance was considered to be a *p*-value < 0.05. All the analyses were stratified by sex.

### 2.5. Ethical Considerations

All information provided by the project partners and beneficiaries was confidential. The participation of the target groups was voluntary and required a signed informed consent document from the school directors, parents, and students. All the project’s procedures and goals were explained in detail to ensure that potential participants, their parents, and teachers were well informed and did not feel forced into giving their consent. Actions were implemented with professionalism, teamwork, proximity, availability, and flexibility. In addition, the coordinator institution (University of Alicante, Spain) and all partners ensured that all individuals working in the project in contact with children had no prior convictions and sanctions and ensured that everyone adopt codes of good conduct and good praxis. The study was conducted according to the guidelines of the Declaration of Helsinki and approved by Ethics Committee of our universities. More details about our Child Protection Policy were described elsewhere [19].

## 3. Results

The final sample was made up of 1008 secondary school students who affirmed having or having had a dating relationship. The average age was 14.3 years (standard deviation, SD = 1.5). More than half (56.1%) of our sample was girls. Girls reported lower social support in all the dimensions (*p* < 0.001). They also reported lower benevolent sexism (BS) and hostile sexism (HS) than boys (*p* < 0.001) and lower violent behaviour in both dimensions of the MVQ, which refer to machismo and acceptability of violence (*p* < 0.001) (Table 1).

The global prevalence of DV victimization was significantly greater among girls than boys (34.1% vs. 26.7%; *p* = 0.012). These differences were also found for the prevalence of exposure to psychological violence involving control and/or fear (girls: 28.1%, boys: 21.0%, *p* = 0.011). There were no statistically significant differences by sex in prevalence of physical and/or sexual violence, although the proportion registered among girls was greater (girls: 16.0%, boys: 11.8%, *p* = 0.069) (Figure 1).

The prevalence of DV in both girls and boys was greater for those over age 15 (girls: 48.5% *p* < 0.001; boys: 35.9% *p* = 0.019) than for younger ages. It was also greater in boys and in girls that reported having suffered family violence in childhood, those who had lower social support from schoolteachers and parents, and those with higher scores for both HS and machismo. Furthermore, in girls the prevalence of DV was greater among those who reported their fathers had non-paid work (51.9%; *p* = 0.004), had lower average social support from a close friend (*p* = 0.002), greater acceptability of violence and lower capacity for problem resolution (*p* < 0.001) (Table 2).

Table 3 shows the variables associated with dating violence in girls. The girls whose father did not have paid employment presented a 39% greater probability of suffering from dating violence than those whose fathers had paid work (*p* = 0.024). Furthermore, girls who had suffered abuse in childhood had a 69% greater chance of having experienced dating violence than girls without abuse in childhood (*p* < 0.001). Also, there was a greater probability of experiencing dating violence in girls with greater BS [PR (CI 95%): 1.01 (1.00–1.03)] and with greater machismo [PR (CI 95%): 1.02 (1.00–1.05)].

Table 4 shows the variables associated with dating violence in boys. Boys who experienced abuse in childhood had 48% greater chance of having experienced dating violence than boys without abuse in childhood (*p* = 0.018). The likelihood of DV decreased in boys who reported higher levels of social support from parents [PR (CI 95%): 0.97 (0.96–0.99)]. There was a greater probability of dating violence for high scores on HS (*p* = 0.019), and higher levels of machismo (*p* < 0.001). The probability of male DV victimization decreases with higher levels of violence acceptance (*p* = 0.009).

## 4. Discussion

Nearly one third of our female sample and one quarter of the male sample declared having been exposed to different forms of violence in dating relationships. The prevalence found in this study is noteworthy, considering the young age of the sample. DV likelihood seems to increase with age in both girls and boys, and with lower household socioeconomic circumstances in the case of girls. Previous abuse during childhood was shown to be the main factor associated with DV victimization in both gender groups, as well as sexism (hostile and/or benevolent) and attitudes of machismo. Adolescents’ relationships with family, friends and teachers, measured through the social support scale, seem to be relevant external assets that help them cope with DV, regardless of their social circumstances.

DV victimization is present among both female and male adolescents. This pattern of adolescents’ exposure to violence and behaviour differs from that observed among adults, with women experiencing more victimization than men. All forms of DV victimization (physical/sexual and psychological) are, nevertheless, more prevalent among girls than boys, as it has been observed in previous studies [26]. Girls who reported family socioeconomic disadvantage showed an increased risk of violence and victimization. Prior research has suggested that family socioeconomic status and financial strain increases the level of perceived stress and family disorganization, which may in turn increase the level of violence and victimization [27,28].

It is noteworthy that the rates of physical-sexual violence found were significantly higher among both sexes than what has been observed among adult women in Europe [1]. As previously mentioned, these forms of violence tend to increase when adolescents are between 14–15 years old and decrease later in adulthood, probably due to the growing capability of young people to recognize the social implications of their acts and the presence of other resources to resolve conflicts and approach dating partners [26].

In our study, social support from parents and teachers appears to be a protective factor against DV in both boys and girls. These factors (external assets) have been identified as promoters of more positive and healthy development in adolescents [29] because they also promote the development of the personal skills needed to deal with risky behaviors [30]. Providing knowledge and increasing their ability to recognize potentially violent situations and their ability to use social support or other protective resources to prevent physical-sexual violence are increasingly included in European school’s curriculums [31].

In this study, DV victimization among girls and boys was also associated with their previous experiences of direct or indirect violence during childhood. As has been previously suggested, children exposed to patterns of disadvantage such as violence and aggression fail to develop personal and social skills to regulate emotions and to build rewarding relationship with others [32]. Research has suggested that a violent family culture may provide youth a biased model of interpersonal interaction, which normalizes aggressive behaviours. Boys and girls may learn that physical and verbal coercion are adequate and acceptable strategies for changing someone else’s behaviour and solving conflicts in their dating relationships [33]. However, the inter-generational transmission of violent behaviour in girls and boys, and for boys specifically, has not been confirmed in other studies [34]. To interpret this association, other potential stressors related to adolescents’ social circumstances and personal skills should be considered [3].

Machismo and sexism were also associated with an increased likelihood of DV in both girls and boys, as has been previously observed in other studies [35]. More specifically, benevolent sexism was associated with increased DV victimization among girls, and machismo and hostile sexism was associated with DV victimization among boys. These associations could be partially due to the way that sexism and machismo reinforce traditional gender stereotypes. They also reinforce adolescents’ acceptability of male dominance [36], which serves to minimize the experience of dating violence and its consequences [37]. In the case of girls, greater DV victimization was associated with benevolent sexism, instead of hostile sexism as in the case of boys. This might be because the behaviours related to benevolent sexism could be more acceptable among girls as they relate to ideas about “protecting” women [15,38].

In addition, we observed that there was a decrease in the likelihood of DV victimization among boys with higher levels of violence acceptance. Previous studies show that boys (rather, than girls) hold the belief that the use of aggression toward the partner is justified under certain circumstances [39]. This is an element in the stereotype of male strength and masculinity. However, as has been previously seen among adults in the research, a high degree of acceptance of violence can reduce the readiness to identify certain circumstances as violent acts that they are exposed to [40]. Educational interventions that work with myths and stereotypes associated with a higher acceptance of DV are needed as well as other type of interventions such as public campaigns in social media.

In interpreting our results, it is necessary to consider that although our sample size was calculated to have sufficient statistical power to analyse our Lights4violence educational intervention results, it was too small to make inferences about the population in the targeted cities and countries. The factors associated with DV that we identified in this study are limited to those for which we gathered enough participant responses for the purposes of the analyses. We could not include information about parents’ education level in our analysis, because many students reported that they did not know this information about their parents. We did not collect data about previous history of mental health problems. Also, outcome measures were gathered using self-reported data, and it is possible that some adolescents may have underestimated or exaggerated their responses. Future studies should include additional evidence that uses teacher and family reports in order to triangulate responses between students, teachers and educators. Finally, the cross-sectional design of our study also limits our interpretations of the direction of the identified associations.

Despite these limitations, this study provided an approximation to how sociodemographic circumstances, the exposure to other forms of violence and the presence or absence of personal and external protective assets against violence influence DV victimization among adolescents. The results reinforce the role of sexism and machismo, as well as previous exposure to family violence, in the increasing likelihood of DV victimization. Our results suggest the importance of carrying out psychosocial interventions targeted at challenging socially accepted norms and attitudes. They also suggest the need to increase social support from parents and teachers and to promote constructive communication among boys and girls in dating relationships in different setting such as schools and social media.

## 5. Conclusions

According to the alarming prevalence registered in this study among adolescents and the identified associated factors, it can be said that population-based interventions which may be addressed to all young girls and boys are needed as well as those that specially target adolescents with prior or current experiences of other forms of violence. Future DV prevention school programs should be dedicated not only to older teenagers but also to younger teens and children of elementary school ages as has been also stated before [41]. Further research is needed to assess if such programs can serve as anticipatory guidance to weaken the possibility of becoming a victim or perpetrator of physical/sexual violence at the older ages of 14–15 years.

## Figures and Tables

**Figure 1 ijerph-18-01414-f001:**
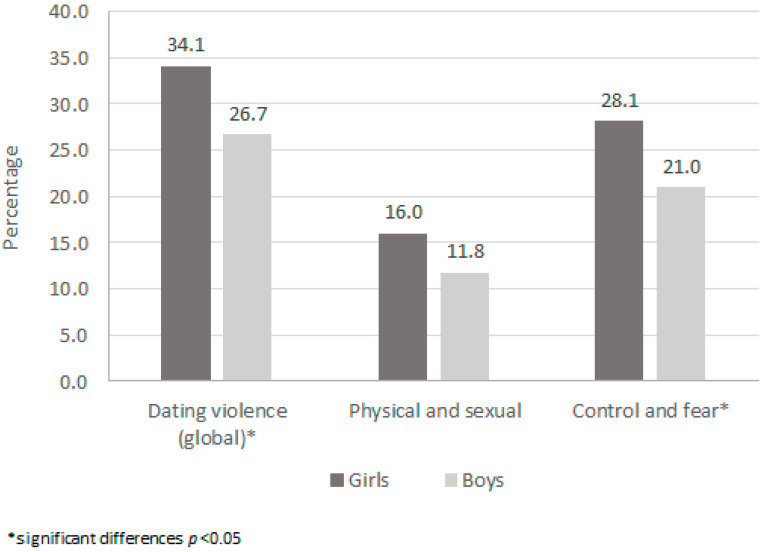
Dating Violence Prevalence by Type and Sex Among Adolescents Ages 13–16 From Alicante, Spain (*n* = 255), Rome, Italy (*n* = 285), Iasi, Romania (*n* = 343), Matosinhos, Portugal (*n* = 259), Poznan, Poland (*n* = 190) and Cardiff, UK (*n* = 204), 2018–2019.

**Table 1 ijerph-18-01414-t001:** Main Characteristics of the Baseline Sample of the Lights4Violence Project from Alicante, Spain (*n* = 255), Rome, Italy (*n* = 285), Iasi, Romania (*n* = 343), Matosinhos, Portugal (*n* = 259), Poznan, Poland (*n* = 190) and Cardiff, UK (*n* = 204), 2018–2019.

Variable	Girls (*n* = 555)	Boys (*n* = 435)	*p*-Value ^a^
	*n* (%)	*n* (%)	
**Father’s employment**			0.844
Non-paid work	54 (10.5)	41 (10.4)	
Paid work	462 (89.5)	352 (89.6)	
**Physical and sexual abuse in childhood**			0.777
Yes	119 (21.7)	96 (22.8)	
No	430 (78.3)	325 (77.2)	
	Mean (SD ^c^)	Mean (SD ^c^)	*p*-value ^b^
**Age**	14.4 (1.5)	14.2 (1.4)	0.035
**Social Support**			
Teacher	48.8 (12.6)	51.7 (12.8)	<0.001
Parents	51.0 (12.3)	55.1 (10.9)	<0.001
Close friend	60.4 (10.1)	57.0 (12.3)	<0.001
**ASI ^d^-Benevolent Sexism**	27.9 (10.9)	30.3 (10.3)	<0.001
**ASI ^d^-Hostile Sexism**	23.9 (10.1)	29.6 (10.0)	<0.001
**MVQ ^e^-Violence Acceptance**	4.6 (3.2)	7.7 (3.1)	<0.001
**MVQ ^e^-Machismo**	6.9 (6.4)	12.0 (8.7)	<0.001
**SPSI-R ^f^-Problem-solving**	58.4 (14.1)	59.1 (12.6)	0.236

^a^ Chi-square test; ^b^ Student *t*-test; ^c^ Standard deviation; ^d^ Ambivalent Sexism Inventory; ^e^ Maudsley Violence Questionnaire; ^f^ Problem-Solving Inventory-Revised Scale.

**Table 2 ijerph-18-01414-t002:** Prevalence of Dating Violence According to Baseline Sample Main Characteristics, Alicante, Spain (*n* = 255), Rome, Italy (*n* = 285), Iasi, Romania (*n* = 343), Matosinhos, Portugal (*n* = 259), Poznan, Poland (*n* = 190) and Cardiff, UK (*n* = 204), 2018–2019.

Variable	Girls			Boys		
	Yes Dating Violence	No Dating Violence	*p*-Value ^b^	Yes Dating Violence	No Dating Violence	*p*-Value ^a^
	*n* (%)	*n* (%)		*n* (%)	*n* (%)	
**Age groups**			<0.001			0.019
≤ 13 years	33 (20.5)	128 (79.5)		28 (19.2)	118 (80.8)	
14–15 years	92 (35.4)	168 (64.6)		57 (28.5)	143 (71.5)	
>15 years	63 (48.5)	67 (51.5)		28 (35.9)	50 (64.1)	
**Father’s employment**			0.004			0.697
Unpaid work	28 (51.9)	26 (48.1)		12 (29.3)	29 (70.7)	
Paid work	149 (32.3)	313 (67.7)		93 (26.4)	259 (73.6)	
**Physical and sexual abuse in childhood**			<0.001			<0.001
Yes	69 (58.0)	50 (42.0)		42 (43.8)	54 (56.3)	
No	117 (27.2)	313 (72.8)		70 (21.5)	255 (78.5)	
	Mean (SD ^c^)	Mean (SD ^c^)	*p*-value ^b^	Mean (SD ^c^)	Mean (SD ^c^)	*p*-value ^b^
**Age**	14.8 (1.3)	14.2 (1.5)	<0.001	14.5 (1.4)	14.1 (1.4)	0.024
**Social Support**						
Teacher	45.5 (11.4)	50.5 (12.9)	<0.001	48.6 (13.1)	52.8 (12.6)	0.003
Parents	47.3 (11.6)	52.9 (12.3)	<0.001	50.4 (11.4)	56.8 (10.2)	<0.001
Close friend	58.6 (11.1)	61.4 (9.4)	0.002	55.1 (13.4)	57.6 (11.8)	0.059
**ASI ^d^-Benevolent Sexism**	28.8 (10.5)	27.4 (11.1)	0.148	30.3 (10.7)	30.3 (10.1)	0.95
**ASI ^d^-Hostile Sexism**	25.3 (10.8)	23.2 (9.6)	0.018	31.5 (10.1)	29.0 (9.9)	0.021
**MVQ ^e^-Violence Acceptance**	5.2 (3.3)	4.3 (3.0)	0.001	7.8 (3.1)	7.7 (3.1)	0.794
**MVQ ^e^-Machismo**	8.8 (7.0)	6.0 (5.9)	<0.001	14.7 (8.8)	11.0 (8.5)	<0.001
**SPSI-R ^f^-Problem-solving**	55.3 (12.9)	59.9 (14.5)	<0.001	57.5 (12.3)	59.7 (12.7)	0.118

^a^ Chi-square test; ^b^ Student *t*-test; ^c^ Standard deviation; ^d^ Ambivalent Sexism Inventory; ^e^ Maudsley Violence Questionnaire; ^f^ Problem-Solving Inventory-Revised Scale.

**Table 3 ijerph-18-01414-t003:** Factors Associated with Dating Violence Among Girls Ages 13–16 From Alicante, Spain (*n* = 255), Rome, Italy (*n* = 285), Iasi, Romania (*n* = 343), Matosinhos, Portugal (*n* = 259), Poznan, Poland (*n* = 190) and Cardiff, UK (*n* = 204), 2018–2019 ^a.^

Variable (Reference)	Crude Model				Adjusted Model			
	PR ^b^	CI ^c^ 95%		*p*-Value	PR ^b^	CI ^c^ 95%		*p*-Value
**Age**	1.20	1.10	1.30	<0.001	1.08	0.95	1.23	0.268
**Father’s employment (Paid work)**								
Unpaid work	1.61	1.20	2.15	0.001	1.39	1.05	1.85	0.024
**Physical and sexual abuse in childhood (No)**								
Yes	2.13	1.71	2.65	<0.001	1.69	1.33	2.16	<0.001
**Social Support**								
Teacher	0.98	0.97	0.99	<0.001	0.99	0.98	1.01	0.342
Parents	0.98	0.97	0.99	<0.001	1.00	0.99	1.01	0.741
Close friend	0.98	0.97	0.99	0.001	0.99	0.98	1.01	0.270
**ASI ^d^-Benevolent Sexism**	1.01	1.00	1.02	0.143	1.01	1.00	1.03	0.025
**ASI ^d^-Hostile Sexism**	1.01	1.00	1.03	0.024	1.00	0.99	1.01	0.907
**MVQ ^e^-Violence Acceptance**	1.06	1.02	1.10	0.001	1.00	0.96	1.04	0.924
**MVQ ^e^-Machismo**	1.04	1.02	1.05	<0.001	1.02	1.00	1.05	0.022
**SPSI-R^f^ Problem-solving**	0.99	0.98	0.99	<0.001	0.99	0.98	1.00	0.084

^a^ Models adjusted by country. ^b^ Prevalence ratio; ^c^ Confidence interval at 95% level; ^d^ Ambivalent Sexism Inventory; ^e^ Maudsley Violence Questionnaire; ^f^ Problem-Solving Inventory-Revised Scale.

**Table 4 ijerph-18-01414-t004:** Factors Associated with Dating Violence Among Boys Ages 13–16 From Alicante, Spain (*n* = 255), Rome, Italy (*n* = 285), Iasi, Romania (*n* = 343), Matosinhos, Portugal (*n* = 259), Poznan, Poland (*n* = 190) and Cardiff, UK (*n* = 204), 2018–2019 ^a.^

Variable (Reference)	Crude Model				Adjusted Model			
	PR ^b^	CI ^c^ 95%		*p*-Value	PR ^b^	CI ^c^ 95%		*p*-Value
**Age**	1.14	1.02	1.27	0.019	0.86	0.72	1.02	0.084
**Father’s employment (Paid work)**								
Unpaid work	1.11	0.67	1.84	0.693	0.99	0.61	1.61	0.956
**Physical and sexual abuse in childhood (No)**								
Yes	2.03	1.49	2.76	<0.001	1.48	1.07	2.06	0.018
**Social Support**								
Teacher	0.98	0.97	0.99	0.002	1.00	0.98	1.01	0.779
Parents	0.96	0.95	0.98	<0.001	0.97	0.96	0.99	<0.001
Close friend	0.98	0.97	0.99	0.004	1.01	1.00	1.02	0.212
**ASI ^d^-Benevolent Sexism**	1.00	0.98	1.02	0.951	1.01	0.99	1.02	0.444
**ASI ^d^-Hostile Sexism**	1.02	1.00	1.03	0.020	1.02	1.00	1.04	0.019
**MVQ ^e^-Violence Acceptance**	1.01	0.96	1.06	0.795	0.92	0.87	0.98	0.009
**MVQ ^e^-Machismo**	1.03	1.02	1.05	<0.001	1.04	1.02	1.07	<0.001
**SPSI-R ^f^ Problem-Solving**	0.99	0.98	1.00	0.110	1.00	0.99	1.02	0.594

^a^ Models adjusted by country. ^b^ Prevalence ratio; ^c^ Confidence interval at 95% level; ^d^ Ambivalent Sexism Inventory; ^e^ Maudsley Violence Questionnaire; ^f^ Problem-Solving Inventory-Revised Scale.

## Data Availability

The datasets and material that was produced during the current study is available from the main author on reasonable request that guarantee their use according to the ethical procedures adopted in this project and participants’ informed consent documents content.

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
