# Peer review of "Dating Violence Victimization among Adolescents in Europe: Baseline Results from the Lights4Violence Project"

_ijerph, 2021, doi:10.3390/ijerph18041414_

Round 1
Reviewer 1 Report
Please see attached

Author Response
The paper entitled “Dating Violence Victimization Among Adolescents in Europe: Baseline Results from the Lights4 Violence Project contains new scientific knowledge and covers a relevant topic. However, I have some comments that have to be addressed before it can be considered for publication.
- Authors should consider introducing new relevant information about dating violence and possible causes of violence in romantic relationships (Muñiz-Rivas, Vera, and Povedano-Diaz, 2019; Ortuño-Sierra, Gutierrez, Chocarro, Perez, and Aritio-Solana, 2021; Taquete and Monteiro, 2019)
Thank you for your recommendation. We added the following ones which fit with our study:
[13] Muñiz-Rivas M, Vera M, Povedano-Díaz A. Parental Style, Dating Violence and Gender. Int J Environ Res Public Health. 2019 Jul 30;16(15):2722.
[14] Taquette SR, Monteiro DLM. Causes and consequences of adolescent dating violence: a systematic review. J Inj Violence Res. 2019 Jul;11(2):137-147.
- The participants sections should provide information about the age distribution. We agree with your suggestion and added this information in the current version of the manuscript.
- In addition, did authors check for previous history of mental health problems. It seems pretty likely that out of 1240 participants some of them have or have had mental healthWe didn’t check it but we added this among our limitations.
- problems. Otherwise, this should be noted in the limitations section.
- Auhors report Chrobach’s Alpha for the instruments used but they do not specified the version reliability of the scores should be provided for each country. In addition, I recommend using McDonals’ Omega, providing the critics that Chrobach’s Alpha has received for ordered categorical data.
- Information about internal consistency was removed, since the goal of this paper is not to do a psychometric validation of the scales. In addition, country was used as a control variable, and data are analyzed for the total sample. However, we would like to show you the score for our sample by country:
|
|
Italy |
Poland |
Portugal |
Romania |
Spain |
UK |
|
Student Social Support (SSSS) |
0.956 |
0.950 |
0.963 |
0.961 |
0.953 |
0.969 |
|
Sexism (ASI)
|
0.787 |
0.930 |
0.906 |
0.819 |
0.924 |
0.815 |
|
Violent thinking (MVQ) |
0.895 |
0.902 |
0.936 |
0.920 |
0.922 |
0.947 |
|
Social Problem-Solving Inventory (SPSI-R) |
0.823 |
0.842 |
0.871 |
0.854 |
0.894 |
0.781 |
- Figure 1 could be more relevant if authors include percentages for each country and theAs we explained in limitations, although our sample size was calculated to have sufficient statistical power to analyse our Lights4violence educational intervention results, it was too small to make inferences about the population in the targeted cities and countries. We considered that provide these prevalences by countries may show a biased magnitude of the problem.
- total sample. Same for the comparisons in Tables 1 and 2.
- Finally, the discussion section may benefit from a deeper explanation and discussion ofThank you for your recommendation. We reviewed and added these references as well as indicated more types of interventions in the current version of the manuscript.
- the possible causes of dating violence (see point 1) and then possible intervention (authors talk about educational interventions, but maybe more lines and citations could be relevant)
Reviewer 2 Report
This manuscript addresses a very relevant and subject: dating violence victimization. This is a very important and timely subject and is a potentially significant contribution to our field. In general, the manuscript follow the requirements of the journal and is very well written, organized and structured. The theoretical introduction is globally well and supporting the empirical study. However, there are some aspects that could be improved.
- In the abstract include a phrase that illustrates the practical implications of the study.
- I consider that the objectives of the study should be in a section of its own, for example, present study, identifying general objective and specific objectives
- The procedures need to be better explained, mainly because the sample involves participants under the age of 18. Nothing is said about informed consent, which needs to be clarified as it was collected.
- In discussing the results, the paragraph described in lines 265-267, is somewhat out of context, mainly because the study focuses on victimization.
- In the conclusions the authors refer “Population-based interventions 342 are needed as well as those that specially target adolescents with prior or current experiences of other forms of violence.”. Authors need to discuss more this recommendation
Therefore, in my opinion the manuscript need Minor Revisions
Author Response
This manuscript addresses a very relevant and subject: dating violence victimization. This is a very important and timely subject and is a potentially significant contribution to our field. In general, the manuscript follow the requirements of the journal and is very well written, organized and structured. The theoretical introduction is globally well and supporting the empirical study. However, there are some aspects that could be improved.
- In the abstract include a phrase that illustrates the practical implications of the study.
- OK. We included one of the most important in the current version of the paper.
- I consider that the objectives of the study should be in a section of its own, for example, present study, identifying general objective and specific objectives Ok. In the current version of the manuscript we added this information in a subsection entitled “Ethical Considerations”.
- The procedures need to be better explained, mainly because the sample involves participants under the age of 18. Nothing is said about informed consent, which needs to be clarified as it was collected.
- We agree and we clarified the main objective of the study in a separate paragraph.
- In discussing the results, the paragraph described in lines 265-267, is somewhat out of context, mainly because the study focuses on victimization.
- We agree and we eliminated it in the current version of the manuscript
- In the conclusions the authors refer “Population-based interventions 342 are needed as well as those that specially target adolescents with prior or current experiences of other forms of violence.”. Authors need to discuss more this recommendation
- Ok, we explained in more detailed the recommendation based on our main results.
Reviewer 3 Report
This is an interesting study related to dating violence victimization among adolescents.
This is an interesting study related to dating violence victimization among adolescents.
However, 3 points should be elucidated or be commented.
- The first paragraph of the conclusions is a general statement; not based exactly on your data.
- Do you refer to particular evidence-based school programs? Which ones?
- The second paragraph of the conclusions should be the objective of other similar studies to see if really such programs could weaken the possibility of becoming a victim at middle adolescence.
Author Response
This is an interesting study related to dating violence victimization among adolescents. However, 3 points should be elucidated or be commented.
1.The first paragraph of the conclusions is a general statement; not based exactly on your data.
- We reformulated our conclusions and justified the idea based on our own results
2.Do you refer to particular evidence-based school programs? Which ones?
As it is clarified in the current version, we refer to DV prevention school programs
3.The second paragraph of the conclusions should be the objective of other similar studies to see if really such programs could weaken the possibility of becoming a victim at middle adolescence.
Ok we added this recommendation in the reviewed version of the manuscript